# Problematic Smartphone Usage in Singaporean University Students: An Analysis of Self-Reported Versus Objectively Measured Smartphone Usage Patterns

**DOI:** 10.3390/healthcare11233033

**Published:** 2023-11-24

**Authors:** James Keng Hong Teo, Iris Yue Ling Chionh, Nasharuddin Akmal Bin Shaul Hamed, Christopher Lai

**Affiliations:** Health and Social Science Cluster, Singapore Institute of Technology, 10 Dover Drive, Singapore 138683, Singapore

**Keywords:** problematic smartphone usage, depression, anxiety, stress, perceived smartphone usage and objective smartphone usage

## Abstract

Introduction: Problematic smartphone usage is the excessive usage of the smartphone, leading to addiction symptoms that impair one’s functional status. Self-administered surveys developed to describe the symptoms and measure the risk of problematic smartphone usage have been associated with depressive symptoms, symptoms of anxiety disorder, and perceived stress. However, self-reported smartphone usage can be unreliable, and previous studies have identified a better association between objectively measured smartphone usage and problematic smartphone usage. Methodology: A self-administered survey was used to investigate the relationships between the risk of problematic smartphone usage (SAS–SV) with depressive symptoms (PHQ–9), anxiety disorder symptoms (GAD–7), and perceived stress (PSS) in Singaporean full-time university students. Self-reported screentime and objectively measured screentime were collected to determine if there is any difference between perceived smartphone usage and objective smartphone usage. Results: There was no statistical difference between self-reported and app-measured screentime in the study population. However, there were significant positive correlations between SAS–SV with PHQ–9, GAD–7, and PSS. In the logistic regression model, PHQ–9 was found to be the sole predictor for variances in SAS–SV score in the study population. Conclusion: This study suggests that problematic smartphone usage may potentially related to depressive symptoms, symptoms of anxiety disorder, and greater perceived stress in university students.

## 1. Introduction

Advancements in mobile phone technology gave rise to the invention of the smartphone, merging the functions of both the computer and the mobile phone. The smartphone enables users access to gaming, the Internet, entertainment, information, social networking, and communication with the ease of a touch. The convenience brought upon by the smartphone has become an essential part of our daily lives, evident in the high global penetration rate of 78% [1]. These handheld devices have become so indispensable in our daily routines that their absence may be perceived as almost crippling [2].

The smartphone adoption rate among university students has been particularly high. Many of these young adults have experienced an increase in smartphone usage with the shift of education to the virtual realm [3]. This demographic, having grown up during the technology boom, is highly receptive to new forms of high-tech media such as smartphones, and college students are especially vulnerable to technology overuse as they transition from adolescence to young adulthood alongside ready access to such equipment [4]. With the virtual world within arm’s reach, users rely on social media and the Internet to maintain communication, share information, and continue their education.

In Singapore, the smartphone penetration rate is high at 88%. In addition, for Singaporeans between the ages of 15 and 24 years old, this figure stood at a staggering 100% in 2021 [5]. Problematic smartphone usage (PSU) is on the rise, particularly among adolescents and young adults [6]. PSU is the excessive usage of the smartphone, leading to addiction symptomology with the corresponding impairment in functional status [7]. Undoubtedly, this is a cause of concern as the younger generation are digital natives who are more prone to the adverse effects of smart media as compared to the older generation, and a study has shown an increasing trend in PSU among adolescents and young adults [8].

Numerous studies have documented the association between problematic smartphone usage and its detrimental effects on physical and psychosocial health; for example, musculoskeletal problems and nerve injuries due to repetitive stress [9] and reduced physical activity [10] are just some of the identified physical impacts of smartphone overuse. PSU has also been negatively related to academic performance [11] and has been linked to low self-esteem and decreased work productivity [12,13].

Most notable, however, are the psychological effects of having a higher PSU on this population. PSU has been correlated with greater psychiatric morbidity, particularly in the areas of depression, anxiety, and stress [14]. The academic stress and new responsibilities faced by these undergraduates as they transition into adulthood make them more vulnerable to mental health challenges [15,16]. These students may turn to their smartphones as a form of escapism to temporarily relieve psychological distress and avoid confronting stressful situations, leading to problematic smartphone use if not regulated properly [17]. Eventually, these individuals may begin to neglect other responsibilities in their lives, resulting in greater levels of negative emotions and stress. It is, hence, pertinent to investigate PSU among local university students and establish any relationship with the levels of depression, anxiety, and perceived stress faced by this population as a basis to distinguish those at risk and identify avenues for support where necessary. Extant literature has evaluated the negative effects of problematic smartphone usage and smartphone addiction on adolescents around the world, illustrating its detrimental effects on both the physical and psychosocial wellbeing of this population [18,19,20].

As PSU is not classified as a form of addictive disorder by the Diagnostic and Statistical Manual of Mental Disorders, several diagnostic tools have been developed to describe and measure PSU [21,22]. An example is the Smartphone Addiction Scale—Short Version (SAS–SV), which was developed and validated by [23]. It is a self-administered questionnaire to screen for symptoms of addiction and evaluate the risk of PSU. The psychometric properties of SAS–SV have been validated among university students in America and China [24,25].

With the aid of SAS–SV, previous studies found an association between PSU and poorer mental wellbeing. For instance, Samaha and Hawi. (2016) found that PSU is positively correlated to stress while negatively correlating with academic performance among 300 university students [26]. The PSU was indirectly associated with reduced satisfaction with life via poorer academic performance and an increase in stress. More recently, in China, Zhang et al. (2022) identified a higher prevalence of depression, insomnia, and anxiety among university students with PSU [27].

However, self-administered questionnaires regarding smartphone behavior can be inaccurate due to recall bias. While self-reported data are commonly used to evaluate smartphone usage due to their ability to provide insight into consumer behavior and mindset, the nature of smartphone usage also makes it difficult for accurate recollection; it was found in a study consisting of 95 Korean college students that these participants interacted with their phones more than a hundred times a day on average with each session lasting several minutes [8]. The fact that smartphones can be used almost anytime and anywhere to support a wide variety of tasks also increases the cognitive difficulty of recalling usage behavior and duration [28].

Alternative methods of measuring media usage can be considered instead; there is an increasing availability of applications and features on smartphone platforms that provide a refined, objective representation of mobile usage, such as logged screen time and application usage. Several previous studies have demonstrated that objectively reported problematic smartphone usage was found to be associated with weaknesses in working memory capacity, task switching, inhibitory control latent factors [29], and poorer self-control [30]. Moreover, increased objective smartphone usage and levels of smartphone addiction were also associated with the underestimation of smartphone usage [31,32].

However, various studies in the field of media and technology usage have also delved into the inconsistencies between perceived and objectively measured behaviors; a systematic review of the discrepancies between self-reported and logged digital media use showed that participants are rarely accurate when asked to estimate their digital media consumption, with similar proportions of studies indicating either under- or over-reporting of media usage [33]. Also, higher degrees of inaccuracy between self-reported and logged measurements were directly related to greater levels of depression, loneliness, and lower life satisfaction in the respondents [34]. At the time of the research, there was little to no literature published that comprehensively investigated the relationship between objectively reported smartphone usage and anxiety, depression, and stress in Singaporean university students. Given the predominance of self-reported data in media and psychology studies, the implications of the non-correspondence between perceived and logged media use observed in the current literature are considerable; findings regarding smartphone usage and wellbeing have the potential to foment societal or policy changes and provide avenues for future research examining its potential significance to public health and possible interventions. Therefore, the aims of this study were to determine (1) if there is any discrepancy between reported smartphone usage time and measured smartphone usage time among Singaporean university students and (2) the relationships between symptoms of PSU with symptoms of anxiety, depression, stress perception, and measured smartphone usage.

## 2. Methodology

### 2.1. Study Design and Participants

This is a survey study conducted among Singaporean university students using a 6-part self-administered questionnaire to investigate the smartphone usage pattern and its effect. The inclusion criteria of the present study include full-time university students between the ages of 21 and 30 years old and those using a smartphone that operates on the Apple iOS operating system. The only exclusion criterion in the present study was those participants who were not using a smartphone with the iPhone Screen Time function (non-Apple iOS users). The study focused on Apple iOS users to standardize the categorization of mobile applications by the in-built screentime function. Standardization of the mobile application is harder to achieve for phones operating on Android iOS as it typically relies on third-party applications for mobile application usage tracking, and the in-built tracking function may vary between smartphone brands. The workflow of the present study is summarized in Figure 1.

#### 2.1.1. Part 1—Self-Reported Smartphone Screentime

The participants estimated and indicated their perceived smartphone usage on four categories of smartphone applications: messaging and social networking (examples: Facebook, WhatsApp, Instagram, Telegram, etc.), web browsing (examples: Safari, Google Chrome, etc.), gaming (examples: Genshin Impact, Mobile Legend, etc.), and entertainment (examples: TikTok, Netflix, YouTube, etc.) for the past seven days. Participants were encouraged to complete this part of the survey truthfully without referencing any screentime tracking smartphone application. The time spent was indicated on the survey using a drag bar in hours to a single decimal point. Examples of smartphone applications for each category were also provided in the survey.

#### 2.1.2. Part 2—Smartphone Addiction Risk

The SAS–SV is a ten-item, six-point Likert scale self-rated questionnaire (1: “Strongly Disagree” to 6: “Strongly Agree”) to evaluate the risk of PSU. One example of the item in the questionnaire is as follows: “I have a hard time concentrating in class, while doing assignments, or while working due to smartphone use’’, participants will rate how well they agree with the statement. The total SAS–SV score for each participant was calculated by adding the points from the ten items altogether. The higher the total points obtained for the questionnaire, the higher the risk of PSU.

#### 2.1.3. Part 3—Perceived Stress

The Perceived Stress Scale (PSS) is a 10-item, self-rated, five-point Likert scale (0: “Never” and 4: “Very Often”) developed by Cohen, Kamarck, and Mermelstein (1983) to assess the participant’s perception of stress [35]. Participants rated how frequently life has been experienced as unpredictable, uncontrollable, and overloaded in the past seven days. An example of the item is as follows: “How often have you been upset because of something that happened unexpectedly?” A total score was calculated by adding up the points from all the ten items, and a higher score indicates a higher level of stress faced by participants. The PSS is simple to use and widely studied with acceptable psychometric properties; therefore, it is recommended for clinical practice and clinical research [36,37].

#### 2.1.4. Part 4—Depressive Symptoms

The Patient Health Questionnaire—9 (PHQ–9) is a nine-item, self-rated, 4-point Likert scale (1: “Not at all” to 4: “nearly every day”) that is used to screen for symptoms of depression. Developed by Kroenke et al., the PHQ–9 can be used to aid clinicians in detecting depression and can be used to monitor the severity of depression. Participants rated the frequency of concerns toward the items for the past 7 days. An example of the item is as follows: “How often have you been bothered by any of the following problems? I have little interest or pleasure in doing things.” The points for all items were tallied for each participant, and a higher score indicates greater severity of depression. PHQ–9 is well studied globally and is implemented in many clinical settings [38]. The recommended cut-off points are Tier 1: total points = 0–4 indicate no depressive symptoms; Tier 2: total points 5–9 indicate mild depressive symptoms; Tier 3: total points = 10–14 indicate moderate depressive symptoms; Tier 4: total points = 15–19 indicate moderately severe depressive symptoms; and Tier 5: total points = 20–27 indicate severe depressive symptoms.

#### 2.1.5. Part 5—Anxiety

The General Anxiety Disorder—7 (GAD–7) is a seven-item, self-rated, 4-point Likert scale (1: “Not at all” to 4: “nearly every day”) questionnaire developed by Spitzer et al. (2006) to screen and measure the severity of for anxiety disorders symptoms [39]. Participants rated the frequency of concerns toward the items for the past 7 days. An example of the item is: “How often have you been bothered by any of the following problems? I feel anxious, nervous, or on edge”. The points for all items were tallied, and a higher score indicates the presence of moderate anxiety disorder. Spitzer et al. have validated this scale in primary healthcare settings and recommended a cut-off point of 10. The GAD–7 is a well-studied, efficient, and valid tool for evaluating anxiety disorder [40].

#### 2.1.6. Part 6—Objectively Measured Screentime

This part of the survey was used to represent the objective smartphone usage time measured and was indicated with reference to the usage time measured by the built-in “Screen Time” application in iOS.

### 2.2. Data Analysis

Data were analyzed with IBM SPSS Statistics 28. All variables were checked for normality before performing statistical analysis. The difference between self-reported screentime and objectively measured screentime was not normally distributed, and hence, the Wilcoxon signed-rank test was used to compare the mean differences. Preliminary analysis of the variables was performed using Kolmogorov–Smirnov to establish normality. Normally distributed variables like SAS–SV, objectively measured screentime, PHQ–9, and PSS relationship were investigated with Pearson correlation, whereas Kendall’s Tau correlation was used to investigate the relationship between SAS–SV with non-normally distributed GAD–7 and self-reported screentime. The linear relationship between each pair of correlations was checked (positive linear regression relationships were confirmed in all pairs of variables), and boxplots were used to ensure there were no outliers (nil outliers were found). After checking the assumptions for linear regression, multiple linear regression (backward) analysis was used to determine the independent predictors of SAS–SV.

## 3. Results

### 3.1. Participants

There was a total of 93 participants responded to the survey invitation. A total of 33 of them were excluded as they did not complete the survey, and 7 of them were excluded because they indicated unrealistic measured time; either the time indicated exceeded 168 h per week (total number of hours in a week) or 0 h per week. Finally, 53 participants were included in the study.

### 3.2. Difference between Self-Reported Screentime and Objectively Measured Screentime

Descriptive statistics of the study population (n = 53) are summarized in Table 1. The objectively measured screentime indicates that the participants in the study population spent an average of 57 h per week or approximately 8.2 h per day on their smartphones. Table 2 summarizes the mean differences between self-reported screentime and objectively measured screentime under different categories. Across the categories, four of the screen times were overestimated except for gaming, which was underestimated by 0.5 h per week. However, there was no statistical difference between self-reported screentime and objectively measured screentime.

### 3.3. Correlations between SAS–SV with PHQ–9, GAD–7, PSS, Objectively Measured Screentime, and Self-Reported Screentime

The Cronbach’s alpha values for SAS–SV, PSS, PHQ–9, and GAD–7 were 0.97, 0.57, 0.86, and 0.86, respectively. The correlation analysis is summarized in Table 3. Overall, there were significant positive correlations between SAS–SV with PHQ–9 (r = 0.66, *p* < 0.001) (Figure 2), GAD–7 (r = 0.42, *p* < 0.001) (Figure 3), PSS (r = 0.5, *p* < 0.001) (Figure 4), and objectively measured screentime (r = 0.41, *p* = 0.02) (Figure 5) except for self-reported screentime (*p* = 0.73)

Finally, a multiple linear regression (backward) was used to determine the predictor of SAS–SV (Table 4). SAS–SV was analyzed as the dependent variables with PHQ–9, GAD–7, PSS, and objectively measured screentime entered into the model as independent variables. The result suggests that PHQ–9 and objectively measured screentime were the key predictors of SAS–SV, explaining 52% of variances in SAS–SV in the study population.

## 4. Discussion

### 4.1. Difference between Self-Reported Screentime and Objectively Measured Screentime

The results from our study showed that there is a discrepancy between self-reported smartphone usage time and objectively measured smartphone usage time among university students, although the mean difference was not significant. Participants may have the tendency to overestimate their smartphone usage time except for gaming, which was underestimated. Hence, caution should be practiced for subsequent studies using self-reported smartphone usage.

Most of our results correspond to a previous study conducted among adolescents and young adults in Hong Kong [41]. In their study, participants overestimated their smartphone usage time by 65%, whereas our participants’ mean total smartphone usage was overestimated by approximately 25%. However, our finding was insignificant, this may be due to the smaller sample size in our study and the dissimilar study population. Furthermore, the Hong Kong study design differs in that the reported screentime was computed as five school-day usage plus two-time holiday usage and had an upper limit of 84 h, while our study asked participants to estimate their usage times for the past seven days and did not set an upper limit.

Our participants had better accuracy in estimating their time spent on gaming (the ideal discrepancies to be as close to zero as possible). This can be explained by the high number of participants that have zero hours spent per week for both self-reported and objectively measured screen time, suggesting that the smartphone is not the preferred mode for gaming in our study population or that our sampled population does not have the habit of gaming. An alternate explanation for the underestimation is that gaming time is easier to recall than other smartphone applications as it is used less frequently, and when used, the session lasts longer [41].

In Taiwan, Lin et al. (2015) termed the underestimation of smartphone usage time as time distortion, a symptom seen in people with Internet addiction. Their study identifies an association between time distortion and their self-designed PSU model, suggesting that time distortion can be used to diagnose PSU [31].

As the scope of the study did not examine the relationship between the accuracy of smartphone usage time estimation with PSU and did not collect data on smartphone application usage frequency, we recommend future studies to examine the reason gaming time is much more accurately predicted and the if underestimation of smartphone usage is related to smartphone addiction.

### 4.2. The Correlations between SAS–SV with PHQ–9, GAD–7, PSS, and Objectively Measured Screentime

This study found significant moderate positive correlations between symptoms of PSU with depressive symptoms and perceived stress and a weak positive correlation with symptoms of anxiety disorder in Singaporean university students. This means that the risk of PSU increases with the mental health parameter measured. Likewise, in a recent study conducted among 548 Malaysian medical students, fair positive correlations were also identified between PSU and depression, anxiety, and stress [42].

In particular to the correlations between SAS–SV with PHQ–9 and GAD–7, a recent study conducted among Chinese university students also found a positive correlation between SAS–SV score with GAD–7 and PHQ–9 scores [27]. Furthermore, a stronger positive correlation was also seen in participants who were attending lessons online. The authors elucidated that the coronavirus disease-19 (COVID-19) outbreak has disrupted students’ normal lives and learning patterns. To reconcile with their negative emotion about the changes, students seek mental compensation from their smartphones. Indulgence in phone usage may lead to loneliness, social phobia, and aggravation or triggering of depression [43]. The compensatory mechanism may be a possible explanation for the positive correlations between SAS–SV with PHQ–9 and GAD–7 in our sample, as our study was also conducted during the COVID-19 pandemic when there were major changes in the mode of learning.

Positive correlations between SAS–SV and PSS were identified among university students by Samaha et al. (2016). The authors suggested that perceived stress and PSU can precipitate one another [26]. Factors like unsatisfactory grades can trigger an increase in perceived stress, which then increases the risk of PSU on a higher level of stress. Matar Boumosleh and Jaalouk (2017) explained that university students with tense personality traits tend to have high stress levels and a lack of positive coping mechanisms, which can lead to increased susceptibility to PSU [20]. Personality traits and grades could be an explanation for the association between perceived stress and SAS–SV in our sample population and should be examined in future studies. However, the finding should be interpreted with care, as PSS is not a reliable scale for our sample.

From the correlation study, our sample has shown that measured screentime is a better predictor of PSU than reported screentime. Self-reported media usage is not a true reflection of actual media usage, as concluded by Parry et al. (2021) [33]. The systematic review and meta-analysis of 106 papers regarding discrepancies between self-reported and logged digital media usage by Parry et al. (2021) found a modest correlation between logged media and self-reported media [33]. The greater the objectively measured screentime, the greater the SAS–SV score, suggesting that excessive smartphone usage is associated with PSU. The correlation resonated with the findings by Randjelovic et al. (2020), where they identified significant numbers of medical students with PSU and that extensive smartphone usage is positively associated with problematic smartphone usage [44].

In our study population, depressive symptoms emerged as the key predictor of PSU symptoms and risk. The results resonated with findings among Korean and Lebanese university students [20,45]. A study by Stankovic et al. (2021) found that increased depressive symptoms cause smartphone overuse via mediation of stress, anxiety, and sleep quality [45]. The Interaction of Person–Affect–Cognition–Execution model explains that an individual with depression is more prone to extensive smartphone usage. This mechanism can be a valid explanation for our predictive model [46]. Although the current evidence suggests that the presence of PSU symptoms is a result of using smartphones to cope with depression, the relationship is not as straightforward. PSU itself may also worsen mental health. Hence, longitudinal studies should be performed to better understand the impacts of depression on PSU.

### 4.3. Limitation of the Study

There are several limitations in this study. Firstly, the cross-sectional study design is unable to establish any causal relationship between symptoms of depression and PSU risk.

Secondly, the study only recruited tertiary students between the ages of 21 and 30 years old. There was no other demographic data, and only iPhone users were recruited. There was no prior screening for mental disorders among participants. Our sample represents a subset of the Singaporean local university students and may be skewed by the presence of participants with existing mental health disorders like depression and anxiety disorder. Hence, the result should not be generalized to the population of smartphone users.

Thirdly, the psychometric properties of the scale and questionnaires used in the present study were not known to be validated in Singapore; hence, there is limited validity and reliability of our findings.

Lastly, the questionnaire was self-administered and relied on the integrity of the participants.

## 5. Conclusions

The present study suggests a potential discrepancy between self-perceived smartphone usage time and objectively measured usage time and warrants a more robust study to be conducted with a bigger sample size. Moreover, PSU symptoms and their potential association with the risk of depressive symptoms, symptoms of anxiety disorder, and perceived stress were observed in Singaporean university students. Therefore, further study is warranted to monitor the impacts of PSU on mental health and the development of mitigating factors.

## Figures and Tables

**Figure 1 healthcare-11-03033-f001:**
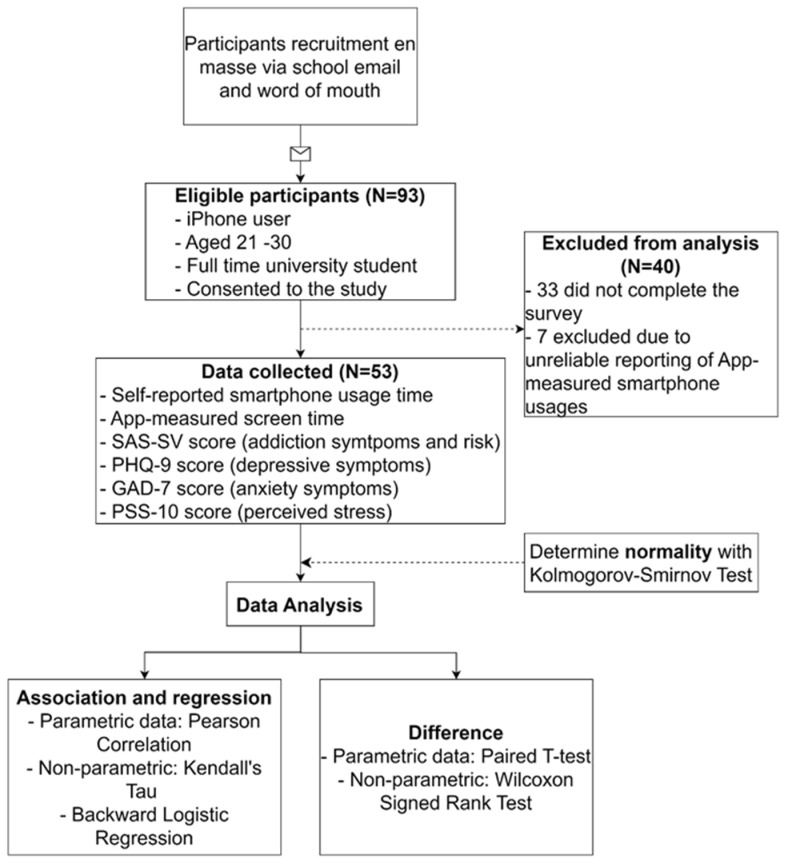
A summary of the workflow of the study.

**Figure 2 healthcare-11-03033-f002:**
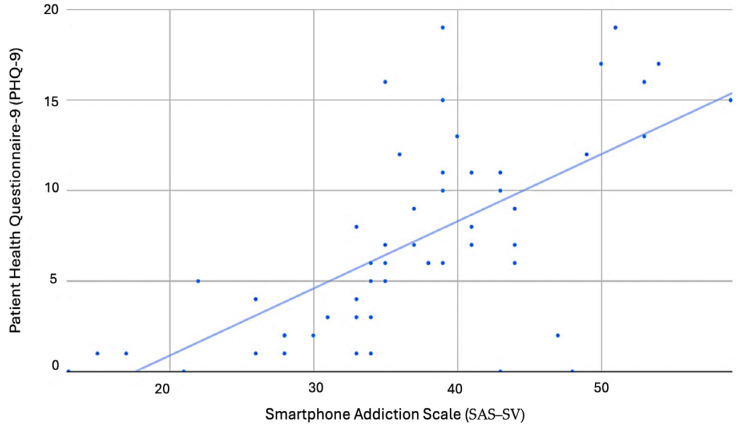
Scatter plot with SAS–SV as the horizontal axis and PHQ–9 as the vertical axis.

**Figure 3 healthcare-11-03033-f003:**
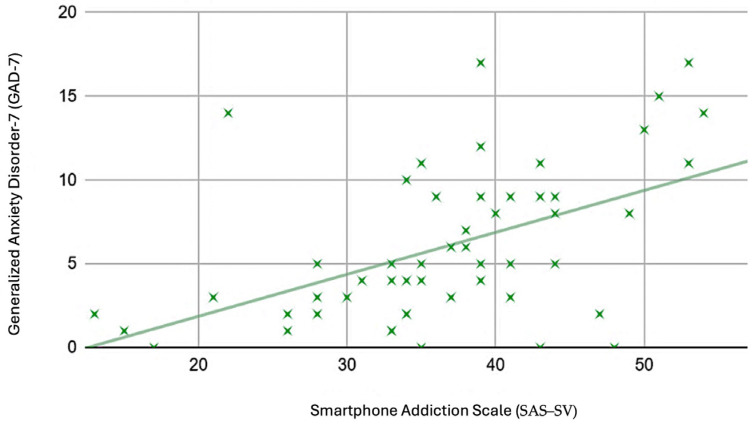
Scatter plot with SAS–SV as the horizontal axis and GAD–7 as the vertical axis.

**Figure 4 healthcare-11-03033-f004:**
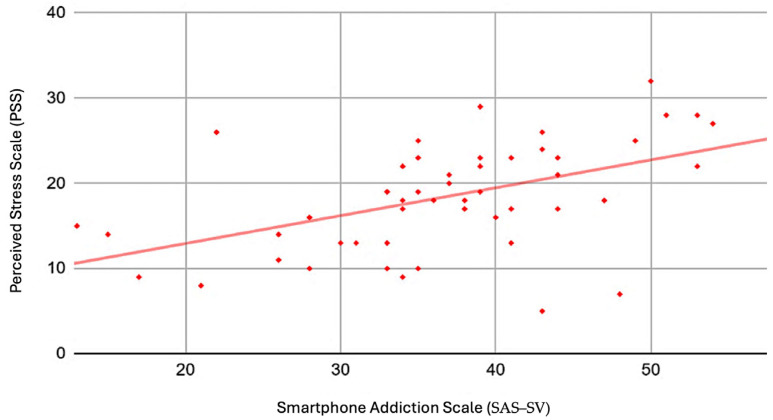
Scatter plot with SAS–SV as the horizontal axis and PSS as the vertical axis.

**Figure 5 healthcare-11-03033-f005:**
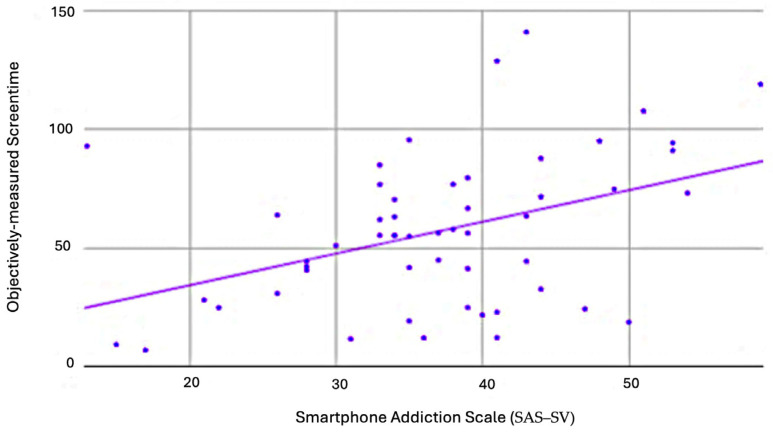
Scatter plot with SAS–SV as the horizontal axis and objectively measured screentime as the vertical axis.

**Table 1 healthcare-11-03033-t001:** A summary of screen time usage in the study population.

	Minimum (Hours)	Maximum (Hours)	Percentiles (Hours)			Self-Reported Statistics Compared to Objectively Measured Statistics (no. of Cases)		
			25th	50th	75th	Over-Estimated	Under-Estimated	Tie
(Self-Reported) Social Networking and Messaging	2.4	111.1	12.8	16.1	26.4	26	24	3
(Objectively measured) Social Networking and Messaging	2.9	50.4	10.4	17.3	26.5			
(Self-Reported) Gaming	0	49.1	0	2.2	14.1	16	15	22
(Objectively measured) Gaming	0	41.7	0	0.5	17.9			
(Self-Reported) Web Browsing	0.5	120.3	5.0	7.6	18.5			
(Objectively measured) Web Browsing	0	43.9	2.7	7.2	12.2	30	22	1
(Self-Reported) Entertainment	3.1	122.8	10.5	14.4	20.9			
(Objectively measured) Entertainment	0	57.4	5.6	21.1	27.4	22	30	1
(Self-Reported) Total Screentime	6.8	141.1	29.4	55.4	76.9			
(Objectively measured) Total Screentime	8.7	317.9	35.8	49.8	80.9	25	28	0

**Table 2 healthcare-11-03033-t002:** Comparison between objectively measured screentime and self-reported screentime in the study population.

Category	Self-Reported Screentime (Hours), Mean (SD)	Objectively Measured Screentime (Hours), Mean (SD)	Mean Difference	*p*-Value
Social Networking and Messaging	25.4 (24.3)	19.5 (11.7)	5.9	*p* = 0.63
Web Browsing Screentime	18.3 (24.8)	9.3 (9.62)	9.0	*p* = 0.64
Gaming Screentime	8.1 (11.9)	8.6 (12.3)	−0.5	*p* = 0.40
Entertainment Screentime	20.4 (21.1)	19.7 (13.9)	0.7	*p* = 0.34
Total	72.1 (59.9)	57.0 (31.8)	15.1	*p* = 0.57

**Table 3 healthcare-11-03033-t003:** Correlations between SAS–SV with PHQ–9, GAD–7, and PSS.

Category	SAS–SV	
	r (Correlation)	*p*-Value
PHQ–9	0.66	<0.001
GAD–7 *	0.42	<0.001
PSS	0.50	<0.001
Self-Reported Screentime *	−0.03	0.73
Objectively Measured Screentime	0.41	0.002

* Correlated to SAS–SV by Kendall’s Tau.

**Table 4 healthcare-11-03033-t004:** Multiple linear regression (backward) analysis with SAS–SV as the dependent variable.

Model		Unstandardized Coefficients		Standardized Coefficients	t	ANOVA		Adjusted R-Square
		B	Std. Error	Beta		F	*p*-Value	
	GAD7	1.11	0.25	0.53	4.44	19.7	<0.001	0.27
	PSS	0.77	0.19	0.5	5.13	17.0	<0.001	0.24
1						14.1	<0.001	0.5
	PSS	−0.14	0.30	−0.09	−0.47			
	GAD7	0.12	0.41	0.06	0.29			
	PHQ9 Objectively Measured Screentime	1.14 0.99	0.33 0.03	0.64 0.32	0.64 3.24			
2						19.1	<0.001	0.51
	PSS	−0.1	0.25	−0.63	−0.38			
	PHQ9 Objectively Measured Screentime	1.18 0.1	0.3 0.0	0.67 0.32	4.11 3.26			
3						29.1	<0.001	0.52
	PHQ9 Objectively Measured Screentime	1.09 0.1	0.17 0.03	0.61 0.32	6.32 3.32			

## Data Availability

The datasets generated during and/or analyzed during the current study are available from the corresponding author on reasonable request.

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
