# Peer review of "Problematic Smartphone Usage in Singaporean University Students: An Analysis of Self-Reported Versus Objectively Measured Smartphone Usage Patterns"

_healthcare, 2023, doi:10.3390/healthcare11233033_

Round 1

Reviewer 1 Report

Comments and Suggestions for Authors

The evaluated manuscript primarily examines the relationship between estimated and actual smartphone usage time, while also assessing the connection between smartphone addiction and conditions such as anxiety, depression, and stress. The article's topic is highly relevant, and the results obtained from both subjective and objective data on smartphone usage may offer valuable insights for future studies. Nevertheless, I believe that the manuscript has several deficiencies that require correction:

Title and abstract:

Given that this is not a psychometric study, I believe it's not justified to use the names of the tests as keywords. Instead, it would be more appropriate to use broader keywords such as 'stress' and 'anxiety,' for instance.

Introduction:

Regarding the paragraph on possible discrepancies between reported smartphone usage and measured usage time: discussing the risk of musculoskeletal harm in this context does seem somewhat unrelated. Instead, it may be more appropriate to introduce a new paragraph in the introduction that explores the relationship between smartphone addiction and its impacts on both psychological and physical well-being. Additionally, it would be beneficial to include a review of the meta-analysis conducted by Parry et al. (2021) on the disparities between logged and self-reported digital media use, complementing the previously mentioned studies: Parry, D. A., Davidson, B. I., Sewall, C. J., Fisher, J. T., Mieczkowski, H., & Quintana, D. S. (2021). A systematic review and meta-analysis of discrepancies between logged and self-reported digital media use. Nature Human Behaviour, 5(11), 1535-1547.

You can move the paragraph on smartphone usage rates in Singapore to follow the first paragraph discussing global usage rates for a more coherent structure.

Methodology:

Could you clarify why the study exclusively focused on Apple iOS users? Providing a brief explanation for this choice would help enhance the understanding of the study's scope and purpose.

It would be beneficial to include an assessment of the tests' reliability, such as Cronbach's alpha or McDonald's omega, to gauge the consistency and accuracy of the measurements. Additionally, introducing an example item from the various tests used could illustrate the nature of the assessments.

While the study workflow mentions the examination of the normality of the variables, it does not explicitly state whether or not the variables meet the assumption of normality. Given that both parametric and nonparametric tests are employed, it is important to specify which variables meet the normality assumption and which do not.

In the section on data analysis, it is mentioned that backward multiple linear regression was performed, while the results indicate a backward logistic regression. This discrepancy should be clarified for accuracy and consistency.

Results:

Please add a note explaining the significance of the asterisk (*) next to GAD-7 in Table 3 to clarify its meaning.

Although it is mentioned in the text that there are no statistically significant differences between self-reported screentime and objectively-measured screentime, the Wilcoxon-Signed Rank Test data should be presented in Table 2 for clarity.

There is no comment regarding whether the assumptions for the regression analysis performed have been tested. It would be beneficial to include information on whether the assumptions were tested and, if possible, their outcomes.

It's understood that the cut-off points proposed by the authors of the SAS-SV were used for the logistic regression analysis. However, for greater clarity, it would be helpful to explicitly state that these cut-off points were employed in the analysis.

Discussion:

In this section, significant attention is devoted to the presence of discrepancies between actual and estimated time spent on smartphone usage. However, it's important to note that, since the results did not reach statistical significance, it's advisable to exercise caution in drawing firm conclusions from these findings.

Reviewer 2 Report

Comments and Suggestions for Authors

This is an interesting study on smartphone addiction awareness in Singapore university students and the potential implications. The paper is timely and I agree that it may contribute well to the literature. I have several comments to improve the manuscript further:

1. The introduction seems to be lacking a clear transition between the prevalence of smartphone usage and its potential negative effects. Consider adding a sentence that bridges these concepts more naturally.

2. The authors should consider to rename smartphone addiction to problematic smartphone given that the term addiction often comes with significant stigma and avoid the risk of overphatologizing the issue. "Problematic smartphone use" more accurately describes a range of behaviors from mild to severe. It captures those who might not be clinically 'addicted' but still face challenges due to their smartphone usage. That is why many researchers and clinicians are moving towards using the term "problematic smartphone use" to capture the essence of the behavior without resorting to the medicalized language of "addiction". Relevant paper: Yu, S., & Sussman, S. (2020). Does smartphone addiction fall on a continuum of addictive behaviors?. International journal of environmental research and public health, 17(2), 422.

2. The authors argued that there is little or no literature published investigating implications of smartphone usage in Singapore university students. Based on my quick search, I found that the statement is misleading. In fact, there are quite a few recent studies in Singapore. The authors should provide a more balanced view on literature and engage more with past relevant research more thoroughly. Several papers on Singapore university students that should be considered: 1. Smartphone use and daily cognitive failures: A critical examination using a daily diary approach with objective smartphone measures. (2023). British Journal of Psychology, 114(1), 70-85. 2. Nuanced relationships between indices of smartphone use and psychological distress: distinguishing problematic smartphone use, phone checking, and screen time. (2023). Behaviour & Information Technology, 1-14. 3. Problematic smartphone usage, objective smartphone engagement, and executive functions: A latent variable analysis. (2023). Attention, Perception, & Psychophysics, 1-16. It will be great to discuss these papers and further highlight the novelty of the current study.

3. Consistency in the format of introducing each scale/tool: When introducing tools like PSS, PHQ-9, and GAD-7, it would be coherent to follow a consistent pattern, e.g., the number of items, the type of Likert scale, followed by the purpose and interpretation of the scale.

4. For clarity, it would be beneficial to detail the method for the objective measurement using the iOS “Screen Time” application. Specifically, how students reported this, and how the team ensured accuracy.

5. It may be beneficial to include any exclusion criteria if they exist. For instance, were students with prior mental health disorders excluded?

6. If any controls or covariates were included in the regression analysis, it might be helpful to list them.

7. I found many typos and grammatical issues in the manuscript. The authors should consider to do a proper proofreading. For example: "This is a survey study conducted amongst the Singapore university students..." should preferably be "This is a survey study conducted among Singapore university students..." "Advancement in mobile phone technology give rise..." should be "Advancement in mobile phone technology gives rise..."

8. In the limitation section, it is important to highlight potential reverse causation in the study. It is possible that higher level of stress and depression is the antecedent rather than outcome of smartphone use.

Author Response

Please see the attachment, thanks

Round 2

Reviewer 1 Report

Comments and Suggestions for Authors

In general, all suggestions have been appropriately addressed.

However, the authors mention that they cannot present Cronbach's alpha or McDonald's Omega because it is a cross-sectional study, incorrectly stating that these statistics assess test-retest reliability. In fact, these tests evaluate the internal consistency of the measures and can be obtained with only one measure per subject.

Table 2 should display the precise p-values rather than indicating solely whether they are greater than 0.05.

Reviewer 2 Report

Comments and Suggestions for Authors

I appreciate and commend the authors' efforts in revising the manuscript. There is a big improvement from the previous version. I believe the current paper will have good implications and contributions to the literature.

I still have a few comments to improve the manuscript further:

1. First, the authors made a good argument regarding the discrepancy between self-reported and logged measurements for smartphone usage. I also agree that self-reported smartphone usage can be unreliable. If that is the clear, it is unclear why the current study did not test the association between apps-measured smartphone usage and the emotional distress outcomes.

2. Given that the scarcity of research in Singapore is a central argument in the last paragraph of the introduction, it would be beneficial to broaden the literature review on studies that focus on objectively reported smartphone usage within the country.

3. I recommend that the authors consider revising the title to eliminate any implication of causality, especially since the word "effects" may suggest a cause-and-effect relationship that is not supported by the cross-sectional study design
